# Understanding the role of psychological distance in preventing the spread of kauri dieback

Hugh A. N. Benson[1]*, Andrea Grant[2], Nicole Lindsay[3], Donald Hine[1]*

**1** School of Psychology, Speech and Hearing, University of Canterbury, Christchurch, New Zealand,
**2** Scion, Christchurch, New Zealand, **3** Department of Psychology, Massey University, Palmerston North, New Zealand

* hugh.benson@pg.canterbury.ac.nz (HANB); donald.hine@canterbury.ac.nz (DH)

## Abstract

### Background

Kauri dieback is a soil-borne pathogen of the family Phytophthora which is lethal to kauri trees. Despite its risks, residents of New Zealand often do not follow imposed mitigation strategies. In this study we explored the potential impact of three factors on psychological distance to kauri dieback: pro-environmental worldviews, trust in government and physical distance from kauri forests. We also investigated the extent to which previously validated psychological distance measures predicted kauri forest visitors' compliance with boot-cleaning and trail-usage guidelines (behaviours linked to the spread of kauri dieback).

### Methods

A survey assessing beliefs and behaviours related to kauri dieback was completed by a sample of 451 New Zealand residents who had visited a kauri forest in the past four years. Two path analyses were conducted to determine whether the effects of environmental worldview (NEP score), trust in government, and physical distance on boot cleaning and track use compliance behaviours were mediated by psychological distance.

### Results

Direct effects indicated that higher NEP score and closer physical distance significantly reduced psychological distance, but trust in government did not. Closer psychological distance also significantly improved self-reported track use and boot cleaning behaviours. Indirect effects indicated that psychological distance significantly mediated the effects of worldview, trust and physical distance on boot cleaning and track usage. Several significant direct effects of the exogenous predictors on the

**Data availability statement:** Data cannot be shared publicly because of restrictions imposed by the Human Research Ethics Committee at the University of Canterbury. These restrictions are imposed because of a descrpency between two information sheets provided to participants of the study. The first information sheet provided did not include a statement mentioning that the anonymised data may be made available to other researchers. While the ethics statement within the survey did state that data may be shared on an online platform. Due to this descrepency, the UC Human Research Ethics Committee has decided upon the more conservative interpretation and restricted online publishing. Despite this, data requests can be made to the Human Research Ethics Committee at human-ethics@canterbury.ac.nz, and should include: * The reference number of this study - HREC 2022/12/LR * Acknowledgement of the restriction. * Research question to be addressed by this dataset. * Hugh Benson's email address, as holder of this dataset - hugh.benson@pg.canterbury.ac.nz * A data management plan, including proposed deletion date. This request should be made prior to 2032, at which time the dataset will be permanently unavailable.

**Funding:** Scion Research is the Crown Research Institute for Forestry in New Zealand. It receives primarily public funding through a variety of sources. Hugh Benson had a prior relationship with Scion Research through his Honour's research supervisor Dr. Andrea Grant (an employee of Scion Research), as well as a Summer Student Research Scholarship of $5,000. Through this relationship Dr. A Grant arranged a Master's Student Scholarship of $25,000 to be paid to Hugh Benson over the course of the degree in regular instalments. This funding enabled Hugh Benson to perform this research and undertake this Master's degree. The only stipulation on this scholarship was that Hugh Benson will attempt to publish his findings, but success in publication was not a stipulation. In this way, the submission of the original manuscript to PLOS ONE fulfilled this obligation. Fulfilment of this stipulation was voluntary by Hugh Benson and failure to fulfil this stipulation would not impose financial repercussions. Prior to this writing, all agreements with Scion Research have been completed and concluded. Dr. Andrea Grant took on a role of co-supervisor for this Master's Thesis, along with Dr. Nicole Lindsay from Massey

compliance behaviours were present after controlling for the mediator, indicative of partial mediation.

## Conclusions

Psychological distance is a reliable predictor of respondents' boot-cleaning and track-use compliance. Interventions to decrease psychological distance may be beneficial for increasing compliance, although the effects were modest and other potential determinants of compliance also require investigation.

## Introduction

Walking through and experiencing native forests is a valued activity for many in Aotearoa/ New Zealand. Yet, seemingly harmless forest strolls, may place kauri forest ecosystems at risk if proper precautions are not taken [1]. The fungus *Phytophthora agathidicida* [2](causing kauri dieback), is deadly to kauri trees and is primarily spread through the transport of contaminated soil or water from an infected area or tree to an uninfected area or tree [3–6]. The most significant vector of transmission is by humans [1,4,6], both within a forest and between forests, and with hundreds of thousands of visitors to kauri forests each year (such as to the very popular tracks in the Waitākere Ranges and Hunua Ranges in Auckland, New Zealand) every single visitor is a potential new vector for the spread of the disease. Currently, the correction of harmful behaviour is required to buffer against two negative outcomes: the continued closure and loss of access to kauri forests for recreationalists, and/or the eventual possibility of collapse of kauri forest ecosystems [3].

The harming of kauri trees is a harming of kauri ora (wellbeing); the entire life and death cycle of kauri trees and the species dependent upon them [7]. Kauri trees have high ecological significance within forests, such as dropping woody litter, leaves, cones and bark. This leaf litter provides important benefits, such as breaking down to release nutrients back into the nutrient cycle, forming an acidic humus which can hinder competing tree species [8], and fostering some native flora, such as the kauri greenhood orchid [3]. Thus, the diseasing of kauri can impact their role in the ecosystem and damage the ecosystem, including disrupting the roles of the plants, birds, and insects, which evolved alongside kauri [8].

### Human behaviour and kauri dieback

Human spreading of spore-contaminated soil into uncontaminated areas is likely the most significant vector of spread for kauri dieback [4], primarily through visitor tracks and informal routes [1]. Rules centred on preventing the transfer of contaminated soil have been implemented in the National Pest Management Plan (2022), or Tiakina Kauri, by the New Zealand Government under the Biosecurity Act (1993) and managed by Biosecurity New Zealand to protect kauri. The 10 rules are [9]: 1. Obligation to report kauri dieback symptoms, 2. Provision of requested information, 3. Restriction on the movement of kauri trees, 4, Persons must adhere to risk management

University. Both of these supervisors provided administrative support, supervision during research gathering, and editing recommendations. Neither Dr. Andrea Grant, nor Dr. Nicole Lindsay, had input into the direction or methodology of the study, this was performed by Prof. Donald Hine of the University of Canterbury, nor did they give editing recommendations that requested alterations to findings, results, or interpretations. The Scholarship was awarded irrespective of findings.

**Competing interests:** The funder (Scion Research) provided support in the form of salaries for author Dr. A Grant and a Master's Student Scholarship for Hugh Benson. Past this financial support Scion Research did not have any additional role in the study design, data collection and analysis, or preparation of the manuscript. Dr. A Grant did provide editing suggestions for consideration by the research team. The specific roles of these authors are articulated in the 'author contributions' section. Suggested edits by editors are available upon reasonable request to Hugh Benson. Any intellectual property created during this research remained the sole property of Hugh Benson. Any prior commercial affiliation does not alter our adherence to PLOS ONE policies on sharing data and materials.

plans, 5. Earthworks may not occur within kauri hygiene zone, 6. Landowners must adhere to stock exclusion notices, 7. Restrictions on the release of animals, 8. Obligation to clean items before entering or exiting kauri forests, 9. Obligation to use cleaning stations, and 10. Compliant use of open tracks and roads in kauri forest. This study is only concerned for rules 8, 9 and 10, and focuses specifically on correct track use and effective boot cleaning. Forest users are required to avoid using closed tracks (signposted and often barred by a fence) and not set foot off the defined track. Along with this, boot cleaning station guidelines specify that footwear should be cleaned both before entering and upon leaving a track, and consists of the removal of all visible dirt by brush, followed by the application of 5% Sterigene solution to kill any remaining active spores [10–12]. All open tracks in both the Waitākere and Hunua Ranges have boot cleaning facilities that are maintained by the Department of Conservation or the Auckland Regional Council. These requirements help prevent forest users from contaminating the track as well as spreading any potential contamination from the track to off-track areas. Although formalised in the 2022 Tiakina Kauri plan, the mitigation strategies of track closure and boot cleaning have been in place since 2018. Despite this, the spread has not stopped, and non-compliance remains a significant risk [13], especially for infecting new areas and thus enabling more natural spread through streams or waterlogged soils [5]. The 2022 regulation will likely have some effect on curtailing the spread of kauri dieback disease, but due to the large geographic area and the millions of visitors to kauri forests yearly, understanding the psychological rationales underpinning the decisions forest users make may aid in the success of encouraging users to follow these proposed mitigation strategies.

## Psychological distance: A mechanism for understanding environmental behaviours

Psychological distance is a framework extension of Construal Level Theory (CLT; explained below) [14] which aims to draw connection between perceived 'distance' (could be described as personal relevance) in which an individual believes they are from an event or its negative outcomes and their likelihood of supporting a specific pertinent belief or action, for example, climate change mitigation policies. Psychological distance is measured based on perceived distance, whereby a lower score indicates the participant feels closer to the issue; that it is more concrete and significant in their mind, with higher scores meaning the issue is less significant and personally relevant.

Psychological distance to environmental threats has been shown to be an important predictor of environmental concern and pro-environmental action, with most studies finding reduced psychological distance is associated with increased pro-environmental intentions and behaviour (PEB) [15]. Although psychological distance studies generally had a climate change focus (with a variety of different outcome variables being evaluated which could represent climate change perceptions), different psychological distance scores resulted in inconsistent effects on the outcome variable [15]. These inconsistencies damage the generalisability of psychological distance from theoretical construct to a practical tool.

Despite these inconsistencies, support for the role of psychological distance in targeted studies focusing on issues with more immediate and visible impact is promising [15]. For example, the increased threat of forest fire (a possible effect of climate change), rather than climate change itself. Unsurprisingly, this indicates that more concrete examples may result in more concrete construal's, and thus lower psychological distance. With this in mind, a more concrete environmental issue, such as kauri dieback, with more concrete behavioural mitigations, both correct boot cleaning and track use, may be an optimal situation in which to employ psychological distance to influence an environmentally deleterious behaviour.

Construal Level Theory provides a useful framework for understanding how significant an issue is to an individual [14]. According to CLT, the extent to which an individual perceives an event as distant is dependent on how the event is mentally construed. Events of greater perceived distance are construed in more abstract terms, as decontextualized and with less specificity in detail, while events of nearer perceived distance are construed in greater detail, and with more concrete, contextualized features. By posing questions aimed at revealing level of construal the answers can be averaged to determine how significant the event is for an individual.

A key element within CLT is psychological distance [14]. The recording of high- and low-level construal can determine how psychologically distant an individual feels from an issue. For example, for those living in or near a kauri forest the effects on and risk to one's own kauri trees, such as the substantial cost for removing a dead tree or the loss of bird habitat (concrete, personal impacts), may further decrease abstraction through the imagined impact being based on a known space, or even specific known trees. Greater abstraction conceptualizations reflect higher psychological distance, whereas more concrete conceptualizations reflect lower distance.

Psychological distance as an extension of CLT is generally conceptualised as a multidimensional construct. Studies have determined four dimensions in psychological distance about climate change: geographic (distance to affected regions), temporal (duration until impact), social (personal level of impact or similarity to impacted people), and uncertainty (doubt as to whether it, or what, will occur; [16–18]. Spence, Poortinga [18] found that individuals can differ in proximity on each dimension, but that most dimensions were moderately to strongly correlated for a context of psychological distance to climate change. This finding was replicated by Jones, Hine [17] and suggests the likely existence of a second order factor structure to which all subcomponents contribute.

### Potential predictors of track users' psychological distance to kauri dieback and PEB

**Pro-environmental behaviours.** Pro-environmental behaviour (PEB) is an individual's behaviour which decreases the negative, or increases the positive, impact one has on the environment [19,20]. Needless to say, it is a broad but simple concept which is applicable from wild dog reporting [21] to turning off the car engine while waiting for a train to pass at a railroad crossing [22]. Encouraging PEB can be through subtracting an environmentally damaging behaviour, such as leaving a defined walking track and in doing so spreading a pathogen, or adding an environmentally beneficial behaviour, such as effective boot cleaning.

In this study three potential predictors of psychological distance were investigated: pro-environmental worldview, trust in government, and physical distance to a kauri dieback infected tree. Pro-environmental worldview was chosen because theoretically an individual with higher eco-centrism will be more emotionally influenced by the potential environmental risk or loss posed by kauri dieback. Trust in government was thought that more trust in the governing bodies proposing or enforcing the guidelines will influence the acceptance and rejection of the likely impact narratives and thus the abstractness or concreteness of these narratives. While physical distance of home address to an infected kauri was thought that this proximity will increase the frequency an individual is exposed to kauri, will increase the perceived impact of kauri dieback, and also increase the practical impact which kauri dieback will have on an individual, such as frequency of experiencing a closed track.

It is important to note the knowledge/action gap which exists in PEB literature [19]. It has been well documented, and even self-evident, that knowledge of an environmental risk does not guarantee action to mitigate this risk. There is no

singular cause of behaviour, but instead a myriad of mixing, conflicting and complementary influences which result in a chosen action. Nor is there perfect consistency, as the choice one day may not be chosen another day. For this reason, a broader theory for behaviour is sought which can consider several influences concurrently.

Employing multiple influences concurrently is the rationale behind using psychological distance. Individuals can vary in the components of psychological distance, temporal distance, social distance, geographic distance, and uncertainty, as well as on the combined psychological distance score. Different inputs can then be added to determine the strength of the psychological effect on how close or distant an individual feels after experiencing this input, and the resultant change in behaviour can be recorded. In this way, the knowledge/action gap can be better explained as the high psychological distance/action gap, where the type or quality of the knowledge may be evaluated as one of the inputs which may reduce psychological distance and through this increase PEB.

This study aims to test three inputs of environmental worldview, trust in government, and physical distance to see if they can influence psychological distance, and if psychological distance can be used to establish an intervention to help mitigate the risks of kauri dieback.

### Pro-environmental worldview

Environmental worldview encompasses a collection of beliefs and values that informs an individual on how the world operates. Generally, it is viewed as a continuum ranging from eco-centric to anthropocentric [23,24], which ironically misses the reality that a healthy environment is ultimately beneficial to humanity, and the thriving of our species [25,26]. Environmental worldviews provide a framework for the perceived 'rightness or wrongness' of behaviour, with pro-environmental worldviews shown to increase PEB and anthropocentric views decreasing PEB [27].

A pro-environmental worldview may lead individuals to perceive environmental issues in more concrete and personally relevant terms (reduced psychological distance). This may be because they are more likely to pay attention to environmental problems, empathize with specific communities, and/or feel a sense of responsibility. Reduced psychological distance, in turn, may make individuals more likely to engage in pro-environmental behaviour. They may participate in conservation efforts, adopt eco-friendly practices, support environmentally responsible policies, or make sustainable consumer choices.

The New Ecological Paradigm (NEP) scale has been widely used to measure anthropocentrism vs ecocentrism worldviews since 1978 [23,24]. The NEP scale measures the extent to which participants adhere to pro-environmental worldviews (ecocentric) over older dominant social paradigm worldviews (anthropocentric) with higher NEP values indicating higher ecocentrism [23].

Debate has occurred as to whether to build upon NEP to explain how it promotes PEB, such as through the mediation of NEP through other variables such as locus of control [27]. Despite the debate, evidence currently shows NEP as an effective tool by itself, and it could not be justified complicating the measure for a possible weak improvement in effect size at this preliminary stage [27]. Although, a follow-up study may wish to consider it.

A similar debate occurred as to replacing NEP with the newer relational value theories [28]. These new theories, such as value belief norms [28], are more robust theories which have the potential to greater inform on the psychological understanding of climate related issues. Unfortunately they did not perform significantly better than NEP for an online panel, and thus the greater complexity was unjustified [28].

### Trust in government

Trust roughly translates into faith that the trusted party is operating in your best interest. In the social sciences trust is often used too broadly, not determining whether the individual trusts a particular authority, group or specific policy, but instead determining whether the individual is of a trusting or distrusting temperament. For example, Cologna, Berthold [29] asked participants whether '[they] trust climate scientists to provide correct information on climate change.' This question

is too unspecific to diagnose trust itself, especially considering scientists in general if asked if they trust the science of their field would likely give a conservative answer. They would likely state a trust of specified researchers, of certain high regarded papers, and a moderate trust of the field in general, but this would not mean that the field is not trustworthy, but that the answer is more complicated than the question.

Kettle and Dow [30] give a useful measure to study trust whereby the break trust down into confidence in abilities and confidence in intentions. It could be argued that confidence in abilities is needed for trust, in the way that low confidence in abilities may erode a high trust in intentions due to little faith that the intentions will be implemented. Yet, confidence in abilities may not influence overall trust if trust in intentions is low, for example, you are unlikely to trust a party who you feel is working against your best interests, even if you do not feel they have the ability to action their plans. Thus, fundamentally, trust in abilities is likely to be irrelevant in low trust situations, but may be relevant in high trust situations. The effect of trust on PEB is often low [29–31], which may point to the social sciences not having a good understanding of trust and how to measure it. Despite this, trust in government likely plays an important role as a heuristic for environmental laymen to actively follow PEB recommendations, and thus warrants its place in studies, despite the limitations of our current tools to capture it.

A link exists between trust of information, rationale, and government, and support for the enacted policy and the requested pro-environmental behaviour. Trust is especially important when there is uncertainty within the debate, and knowledge of the subject is low [31]. A recent study showed how trust in specific regulations moderated the relationship between environmental values and the exhibited behaviours [32]; impacting the expression of environmentally risky behaviours or mitigation behaviours of boating recreationists. The paper also showed how levels of trust in government can activate (or deactivate) these values [32].

Trust is difficult in climate change research, partially due to the difficulty of knowing what underlies measures of trust, which is a wide and complex matter [31]. Other studies have found higher trust in government institutions is correlated with PEB [29,31]. Psychological distance may be able to reincorporate some of this lack of clarity. Trust may be mediated by psychological distance, especially through the uncertainty component (the uncertainty of what the future impacts of an environmental issue may be, and when/if they will occur, and who, if anyone, will be affected), by providing a trustworthy heuristic promoting a suitable PEB. The heuristic shortcuts individuals needing to either figure out a mitigation effort for themselves or being trapped into inaction by the large amount of information required to understand the issue for themselves. People who trust government to provide them with sound advice about the threat of kauri dieback and how to best avoid spreading it are more likely believe what they have been told and to follow the advice.

## Physical distance

A note to begin, geographic distance is a component of psychological distance, which refers to the perceived distance from the issue of kauri dieback. Physical distance is different to geographic distance, and refers to how far away an individual's home address is from an infected kauri tree.

There is general agreement that the effects of climate change will be bad, but these effects will be particularly bad if you are impacted directly. For example, if sea level rise impacts your country, there may be some financial impacts to mitigate the damage, but if your own home is at risk or destroyed, then these impacts will be far greater. Proximity to the issue increases the scale of your potential problem. This has been supported by studies examining how drawing attention to local impacts of climate change can increase the mitigation intentions for those living in the higher risk areas [33,34].

For this reason, physical distance may impact psychological distance, potentially through greater local knowledge of the risks and awareness of the environmental issues. The local nature of the issue may reduce the psychological distance, which in turn can increase the awareness which can drive pro-environmental decision making.

The physical proximity may also enhance the perceived relevance of the threat, such as living in a kauri forest may make the threat of kauri dieback more tangible, as the consequences of even single prominent trees becoming sick can

have an impact on individuals' lives, property and the local environment [35]. The heightened relevance may reduce psychological distance from the issue, making individuals more likely to engage in threat mitigation behaviours and comply with suggested guidelines.

It could also be expected that the inverse is true, that the effect of being more and more geographically distant from kauri forests, reduces the relevance of knowledge and information, increasing psychological distance and reducing the mitigation behaviours.

### The current study

The current study aims to investigate the relationship between psychological distance and behaviours linked to the spread of kauri dieback; specifically, we hypothesise that reduced psychological distance will increase compliance with boot cleaning and track usage guidelines. Next, we expect that low trust in government, high physical distance, and a weak pro-environmental worldview will separately predict greater psychological distance.

Lastly, that the effects of pro-environmental worldview, trust in government, and physical distance, on boot cleaning and track use compliance will be mediated by psychological distance. In particular, we predict that track users who were more trusting in government, held more pro-environmental worldviews, and lived closer to kauri forests will exhibit lower levels of psychological distance to kauri dieback, which in turn will be associated with greater compliance with track use and boot cleaning guidelines.

## Method

### Participants

Two New Zealand samples were collected in two waves from the 15th until the 29th of October 2021 and 15th of February to the 31st July 2022. Participants were informed by written overview of the study on the first page of the survey, and written consent was confirmed through the participant agreeing to participate by selecting 'I agree to participate', refusal ended the survey. Both waves had the inclusion criteria of having walked in the Waitākere and/or Hunua Ranges within the last four years and be aged 18 or over. The participants could live anywhere in New Zealand. Four years was chosen to compensate for the reduction in travel caused by the Covid lockdowns since March 2020. The first sample attained 349 adults through the online research panel Qualtrics. Secondary recruitment was aimed at gathering participants from Titirangi, Auckland, which is a located within a kauri forest suffering from the effects of kauri dieback. Two methods of recruitment made up this Titirangi sample. The first was a mail drop down randomly selected streets in Titirangi, while the second was online social media recruitment using the Titirangi Community Facebook page. These two waves gathered another 102 participants from Titirangi and the immediate surrounding area. This second sample was decided to ensure reasonable representation of those living within a kauri forest, and thus theoretically having a much closer psychological distance compared to those living outside of the forest. The combined total was 451 responses for analysis. Demographic breakdown is presented in Table 1. Ethics has been approved under University of Canterbury Human Research Ethics Committee Low Risk: HREC 2022/12/LR

### Materials

The online survey consisted of sections assessing demographics, psychological distance, self-reported compliance behaviours (boot cleaning and track use behaviours), trust in government, and pro-environmental worldview. Physical location was obtained by requesting participants' postcode.

### Measures

**Demographics.** Common demographic information was requested: sex, age, highest educational achievement, and ethnicity.

**Table 1. Summary of Participant Demographic Information.**

| Demographic | | n (%/100%) | 2018 Census Data %'s(%/100%) | M | SD |
|---|---|---|---|---|---|
| Gender | | n = 451 | | 1.6 | 0.5 |
| | Male (1) | 172 (38.1) | (49.4) | | |
| | Female (2) | 275 (61.0) | (50.6) | | |
| Age | | n = 451 | Total 75.6% (100%) | 40.5 | 16.1 |
| | 18–19 | 24 (5.3) | 1.6 (2.1) | | |
| | 20–24 | 49 (10.9) | 6.8 (9.0) | | |
| | 25–34 | 121 (26.8) | 14.1 (18.7) | | |
| | 35–44 | 100 (22.2) | 12.5 (16.5) | | |
| | 45–54 | 61 (13.5) | 13.4 (17.7) | | |
| | 55–64 | 48 (10.6) | 12.0 (15.9) | | |
| | 65+ | 48 (10.6) | 15.2 (20.1) | | |
| Highest educational attainment | | n = 451 | | 3.7 | 1.1 |
| | Primary School (1) | 1 (0.2) | (18.2) | | |
| | Secondary School (2) | 82 (18.2) | (32.4) | | |
| | Trade Certificate/Diploma (3) | 113 (25.1) | (24.2) | | |
| | Undergraduate Degree (4) | 115 (25.5) | (14.6) | | |
| | Postgraduate/ Professional Degree (5) | 140 (31.0) | (10.2) | | |
| Ethnicity | | n = 468 | Total 111.1% (100%) | | |
| | Pakeha/New Zealand European | 328 (70.1) | 70.2 (63.2) | | |
| | Asian | 70 (15.0) | 15.1 (13.6) | | |
| | Māori | 45 (9.6) | 16.5 (14.9) | | |
| | Pacific Islander | 8 (1.7) | 8.1 (7.3) | | |
| | Other | 17 (3.6) | 1.2 (1.1) | | |

*Notes:* Total sample N = 451, Gender non-binary/other n = 4 (0.9%). Participants who recorded more than one ethnicity were counted in both applicable groups (as per the 2018 NZ Census) n = 29. 2018 Census Data %'s have been adjusted to be percentage of relevant ranges (e.g., 20–24 range 6.8/75.6 = 9.0). Original counts: Māori – 16 (3.5%), Pakeha = 299 (66.3%), Bi-racial Māori/Pakeha = 29 (6.4%).

### Kauri dieback psychological distance

Perceived psychological distance of the effects of kauri dieback on kauri forests was assessed using a new 16-item scale developed for this study. The scale was adapted from Jones, Hine [17] measure of psychological distance from climate change. The scale assessed four dimensions of psychological distance: geographic, temporal, social, and uncertainty. Responses were recorded on a 5-point Likert scale ranging from 1 (strongly disagree) to 5 (strongly agree), 11 items were reverse coded prior to analysis. There was no psychological distance measure for kauri dieback, so the adaptation of a climate change measure was considered appropriate due to the similarities in impact: that impacts will be geographically inconsistent, with some areas being impacted more than others; that the major effects will occur over longer time periods and thus cause and effect may be temporally distant; along with geographic inconsistency, there will be social inconsistencies, such as the personal effect on property owners with kauri trees versus without kauri trees; and uncertainty as to what the ultimate impacts will be. Thus, adaptation of climate change psychological distance questions was considered suitable for the kauri dieback context.

Preliminary factor analytic work, using maximum likelihood extraction and oblimin rotations failed to produce an interpretable multidimensional solution. Thus, following Spence, Poortinga [18], the averaged 16 items were computed into a single unidimensional psychological distance score, showing good internal consistency (*Cronbach's α = .82;* S1 Appendix).

### Self-reported compliance behaviours: boot cleaning and track use behaviour

We assessed two compliance behaviours relevant to reducing the spread of kauri dieback: boot cleaning compliance (boot cleaning) and track use compliance (track use). These are self-reported behaviours, so may differ from actual exhibited behaviours, but due to the difficulty in recording real world behaviours, self-reported was considered to be suitable. All compliance behaviour responses were coded on a 5-point Likert scale which recorded the responses from 1 (Never) to 5 (Always), with reverse coding set up prior to analysis. Boot cleaning was assessed by four items measuring whether 1. boots were cleaned before track use, 2. cleaned after track use, 3. all soil was removed in cleaning, and 4. the Sterigene solution was applied. Reliability analysis conducted on our sample indicated very good internal consistency for boot cleaning (*Cronbach's α* = .87). Track use was assessed by three items measuring whether participants 1. used closed tracks, 2. stayed on the track, and 3. allowed their dog and/or children to leave the tracks (reverse coded). Reliability analysis showed adequate internal consistency for the three items (*Cronbach's α* = .74).

### Trust in government

The Trust in Government scale is made up of four questions adapted from seven questions of the Trust in Institutions scale by Kettle and Dow [30]. Due to no measure existing at the time for trust in institutions relevant or specified to the kauri dieback context, questions were adapted from climate change and were used to determine the level of trust participants had for local government authority responsible for the two kauri forested areas designated as regional park reserves. Despite the low internal reliability for our sample of this adapted scale (*Cronbach's α* = .67), which is not uncommonly low for Trust measures, this scale was included in the analysis as it is hypothesized that trust is important in individuals' receptiveness to communications and thus their comprehension of the issue of kauri dieback. These questions were adapted by the research team and included questions such as '[the local government] has similar priorities and values as I do about kauri dieback.', and '[the local government] has sufficient staff expertise to implement policies about kauri dieback.' Answers were recorded on a 1–5 Likert scale.

### Environmental worldview

A 6-item short version of the New Environmental Paradigm (NEP) scale by Kurisu [36] was used to assess pro-environmental worldview, with higher NEP scores indicating a higher pro-environmental worldview. Similar to other studies using the shortened NEP scale with the New Zealand public, as explored by Balador, Gjerde [37], the measure approached adequate internal reliability with our sample (*Cronbach's α* = .68). Despite approaching adequate internal reliability, it was decided to include the NEP scale in analysis. This was decided due to the NEP generally not mapping well on the New Zealand population [37], but being close enough to inform something about kauri dieback mitigation and psychological distance. In hindsight, the prior debate about relational values should be reconsidered for future studies in the New Zealand to attempt to improve this marginal reliability [28].

### Physical distance

The kauri tree has a limited range in New Zealand, only appearing naturally in the northern third of the country. Thus, determining physical distance is a challenging task, requiring the balance of nuance and clarity. The initial idea was to use longitudinal distance from the geographic kauri ranges, but this only allows for the impact of living outside of the kauri forest geographic range to be measured, which lacks nuance.

In New Zealand postcodes are approximately sequential from north to south. Thus, participant postcodes could be used to generate a physical distance score, at least at the postcode level of resolution. Each participant postcode was mapped and the nearest postcode containing infected kauri was found. A postcode distance score was calculated using the postcode reported by the participant, minus the postcode of the nearest infected kauri tree. The absolute values were used to generate an individual measure of distance from their nearest infected kauri.

An issue arose: how distance would theoretically work in relation to access to kauri forests and exposure to infected kauri. The issue supposes that the effect of physical distance on psychological scores works in a linear fashion, e.g., that a participant who lives 600 km (or 7000 postcodes away, as the case may be) from their nearest infected kauri tree is twice as psychologically distant and has half the forest access, as a participant who lives 300 km from their nearest infected kauri tree. Yet, this is unlikely to reflect reality as it could be hypothesized that someone who lives half the distance is unlikely to care about kauri forests twice as much, thus using postcodes linearly underweights near postcodes. A logarithmic conversion allows the linear measure of distance to be treated exponentially for impact on participants psychological distance. Allowing an overweighing in degrees of nearness, such as someone who lives next to an infected kauri tree, thus experiences this issue daily and may be impacted much more severely than another person who lives even a single postcode away. While under weighing the linear impact of increasing distance, for example for participants living hundreds of kilometres away from an infected kauri the linear effect of an increase in postcode is unlikely to make very little difference in experienced psychological distance. Converting the linear scale of postcode distance to the exponential scale was determined to more accurately reflect how distance is likely to impact individuals.

To create this exponential distance scale (called Physical Distance) the participants postcodes were subtracted from the postcode with the nearest infected kauri tree. Scores were given the absolute value and any scores of zero (same postcode as an infected kauri tree) were given a score of 1.01 (there were no postcode difference scores of one) to allow logarithmic conversion. Then a logarithmic conversion using the value of 2.5 was conducted to provide a physical distance score from 0.1 (in the same postcode) to 9.5 (most distant participant). This value was decided upon to allow the distance to roughly be recorded on a scale of 0–10. Although this logarithmic conversion is more complex and requires more intuitive assumptions on how distance impacts psychological scores, the greater ability to distinguish the impact of physical distance warrants this complexity. Specifically, that those living in versus close to kauri forests will be able to be distinguished, but the distant participants, for example, in the south island of New Zealand who would require a very long drive or a flight to visit a kauri forest will have this distance condensed in their scores, and thus the distant versus the very distance has less impact on their physical distance score.

## Results

### Descriptive statistics and preliminary correlational analyses

Assumptions of normality were tested and all variables breached normality at the $p < .001$ level of Kolmogorov-Smirnov normality test. Checking the histograms and QQ plots, this breach of normality is likely due to a combination of skewness and kurtosis of the variables, coupled with the large sample size. Means and standard deviations can be seen in Table 2. The variables of least concern were NEP (skewness = −0.20, excess kurtosis = −0.82), Trust in Government

**Table 2. Correlation Matrix for Variables.**

| Variable | 1 | 2 | 3 | 4 | 5 | 6 |
|---|---|---|---|---|---|---|
| 1.NEP Score | – | −.29** | −.22** | −.45** | −.25** | .35** |
| 2. Physical Distance | | – | .22** | .38** | −.06 | −.21** |
| 3. Trust in Government | | | – | .04 | .05 | .03 |
| 4. Psychological Distance | | | | – | −.27** | −.34** |
| 5. Boot Cleaning Compliance | | | | | – | .46** |
| 6. Track Use Compliance | | | | | | – |
| Mean | 3.9 | 4.6 | 2.9 | 2.8 | 4.4 | 4.5 |
| Standard Deviation | 0.7 | 3.3 | 0.8 | 0.6 | 0.9 | 0.7 |

Notes: * $p < .05$ level (2-tailed), ** $p < .01$ level (2-tailed), Spearman's *rho* correlations, New Ecological paradigm (NEP), $n = 451$.

(skewness = −0.17, excess kurtosis = −0.42), and Psychological Distance (skewness = −0.57, excess kurtosis = −0.42) which showed acceptable skewness (skewness threshold = 0 +/-1 and excess kurtosis threshold = 0 +/- 1). Two variables showed a ceiling effect as seen by the moderate skew (skewness = 0 +/- 1<=2 and moderate excess kurtosis (kurtosis = 0 +/- 1<=2), Boot Cleaning Complaince (Skewness = −1.83, Kurtosis = 2.95) and Track Use Compliance (Skewness = −1.87, Kurtosis = 3.39), and Physical Distance showed a floor effect (skewness = −0.18, excess kurtosis = −1.49). These moderate violations are understandable due to both the general high level of compliance in kauri forests, that self-reported data is expected to show higher levels of compliance, and that the Titirangi community is based within a kauri forest (giving a high rate for the lowest possible value for Physical Distance). Given the minor to moderate violations of normality, the proposed analysis was still conducted, but results should be interpreted with some caution, especially around behavioural implications.

IBM SPSS Statistics for Windows version 28.0.1.0 [38] was used to analyse the dataset. Means, standard deviations and intercorrelations for all variables are presented in Table 2. On average, participants reported moderate psychological distance to kauri dieback, were slightly distrusting of government, and were moderately to highly ecocentric on the NEP scale.

Physical distance ranged from a minimum value of 0.1, meaning the participant resides within the same postcode as an infected kauri, to a maximum of 9.5, residing at the bottom of the South Island (most distant a New Zealand resident could be from a kauri tree). Most participants did not reside in the same postcode as an infected kauri, with an average physical distance score of 4.6. A quarter of participants lived in the same postcode as an infected kauri (value of 0.1, 25.5%) indicating daily exposure to kauri, while a third (inclusive) lived less than 20 km from an infected kauri (a score of of 1.96 and below, 32.8%), indicating frequent exposure. About a third lived moderately far at between 20 km and 50 km from an infected kauri (between 2.0 to 6.7, 33.1%), indicating less frequent exposure. It should be noted that most of Auckland city (New Zealand's most populas city) sits within a 50 km range of the Waitākere Ranges, a large kauri forest. Above 50 km distance from an infected kauri likely indicates occassional to rare exposure and contained the final third of the sample (scoring above 6.7, 34.3%). It should be noted that all participants had visited a kauri forest within the last four years, as per the inclusion criteria.

As shown in Table 2, the correlations supported the hypothesises that higher NEP scores and closer physical distance would decrease psychological distance, as seen by the significant moderate negative correlation between NEP and psychological distance and the significant moderate positive correlation between physical distance and psychological distance. Our other hypothesis concerning variables impacting psychological distance was not supported, as there was no significant correlation between trust in government and psychological distance. Also in line with the hypothesises, psychological distance was significantly negatively correlated with self-reported compliance to boot cleaning and track use guidelines.

Track use compliance showed a significant moderate positive relationship with NEP score, and a significant weak negative relationship with physical distance. Compliance with boot cleaning guidelines showed a significant weak positive relationship with NEP score.

## Path analysis

Path analysis using IBM SPSS AMOS (28.0.0) by Arbuckle [39] was conducted to test the hypothesis that trust in government, higher NEP scores and physical distance from an infected kauri (Physical Distance) would predict lower levels of psychological distance to kauri dieback (Psychological Distance), which would then be predictive of compliance with both boot cleaning (Boot Cleaning) and track use (Track Use) guidelines. Two separate models were tested, with each predictor variable being tested separately.

Multivariate normality was tested using Mardia's test in RStudio version 2025.05.01 build 513 [40]. Multivariate normality was breached, for the same reasons as the normality assumptions mentioned above, but the research team did not

believe the breaches would impact the model sufficiently to warrant removal of variables, especially considering the large sample size. More conservative thresholds were imposed where possible, such as robust maximum likelihood estimation (for the Track Use, non-saturated, path analysis in Fig 2; using RStudio), and models were compared against the more conservative versions and were found to give no statistically significant difference in results.

## Boot cleaning behaviours path analysis

The first path analysis included paths from all exogenous variables (NEP score, Physical Distance, and Trust in Government) to both Psychological Distance and Boot Cleaning, and from Psychological Distance to Boot Cleaning (Fig 1). All pathways were significant, resulting in a fully saturated model.

As hypothesized, higher NEP scores, higher trust, and lower physical distance predicted lower levels of psychological distance to kauri dieback. In turn, respondents who reported lower psychological distance reported higher levels of boot cleaning compliance. All three indirect pathways from the distal predictors through psychological distance to boot cleaning compliance were statistically significant (see note for Fig 1). Notably, the direct effects of worldview, physical distance and trust in government on boot cleaning compliance all remained statistically significant after controlling for psychological distance, which shows partial mediation, and that these variables effect self-reported boot cleaning compliance directly (or through an uncaptured mediator) and not exclusively through psychological distance. The predictor variables accounted for 32% of variance in psychological distance, but only 11% of variance in Boot Cleaning.

## Path analyses for track use behaviours

The second path analysis included paths from all exogenous variables (NEP score, Physical Distance, and Trust in Government) to both Psychological Distance and Track Use, and from Psychological Distance to Track Use. One non-significant direct path was found between Physical Distance and Track Use, so was excluded from the model. All other paths were significant (Fig 2).

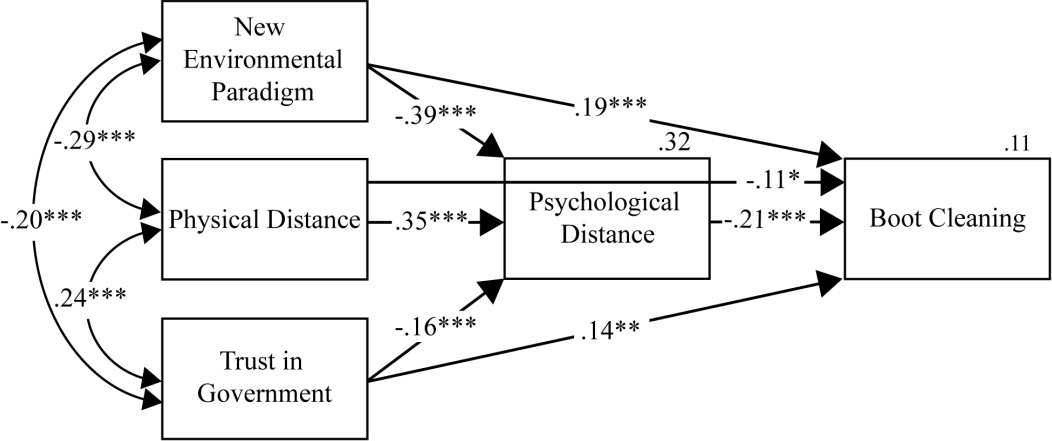

**Fig 1. Path Analysis for the Direct and Indirect Paths of Trust in Government, New Environmental Paradigm and Physical Distance on Boot Cleaning, and Mediation by Psychological Distance.** Notes: The model indicated that higher New Environmental Paradigm scores, higher Trust in Government, and nearer distance from an infected kauri tree (Physical Distance) all predicted closer Psychological Distance and greater compliance to boot cleaning guidelines (Boot Cleaning). Closer Psychological Distance also predicted higher boot cleaning compliance. All three indirect paths from New Environmental Paradigm (β = .12**), Trust in Government (β = .06**), and Physical Distance (β = −.03**), to Boot Cleaning were significant. All path coefficients were standardized. *p < .050, **p < .010, ***p < .001, n = 451.

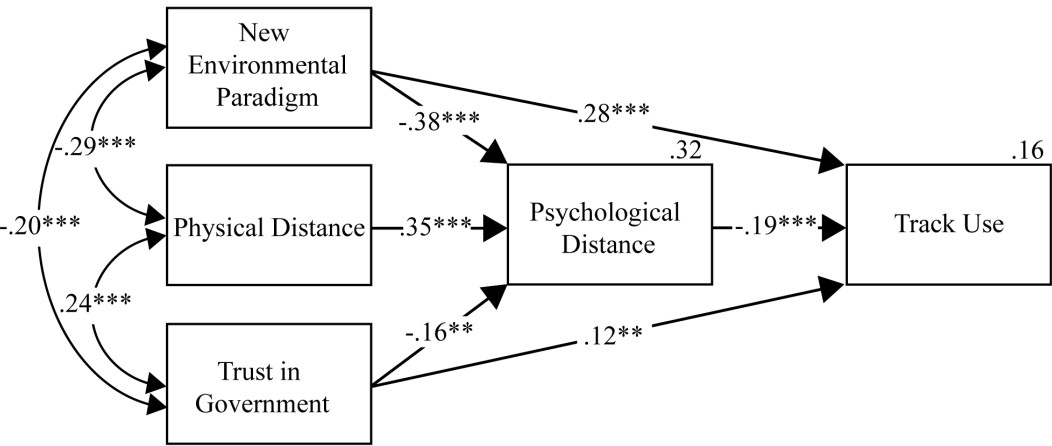

**Fig 2. Path Analysis for the Direct and Indirect Paths of Trust in Government, New Environmental Paradigm and Physical Distance on Track Use, and Mediation by Psychological Distance.** Notes: The model indicated that higher New Environmental Paradigm scores, higher Trust in Government, and nearer distance from an infected kauri tree (Physical Distance) all predicted closer Psychological Distance. New Environmental Paradigm and Trust in Government both predicted greater compliance to track use guidelines (Track Use). Closer Psychological Distance also predicted higher track use compliance. All three indirect paths from New Environmental Paradigm ($\beta = .11^{**}$), Trust in Government ($\beta = .06^{**}$), and Physical Distance ($\beta = -.03^{**}$), to Track Use were significant. All path coefficients were standardized. $^*p < .050$, $^{**}p < .010$, $^{***}p < .001$, $n = 451$.

As hypothesized, stronger pro-environmental worldviews (higher NEP scores), higher trust in government, and lower physical distance predicted lower levels of psychological distance to kauri dieback. In turn, respondents who reported lower psychological distance reported higher levels of boot cleaning compliance. All three indirect pathways from the distal predictors through psychological distance to boot cleaning compliance were statistically significant (see note for Fig 2). Notably, the direct effects of NEP score and trust in government on reported boot cleaning compliance remained statistically significant after controlling for psychological distance. Overall, this pattern of results is consistent with partial mediation, and suggests that other mediational mechanisms, in addition to psychological distance, may be operating. The predictor variables accounted for 32% of variance in psychological distance, and 16% of variance in track use.

## Discussion

This study aimed to extend the current body of psychological distance research on the specific and geographically limited issue of kauri dieback. The first hypothesis was supported in that higher NEP, lower physical distance, and higher trust in government, all successfully predicted lower psychological distance. Lower psychological distance also mediated self-reported compliance with both boot cleaning and track use guidelines. Although, in both cases this was a weak relationship. The second hypothesis was also supported in that all criterion variables were partially mediated by psychological distance, except for physical distance with track use compliance, which shows full mediation by psychological distance due to the significant indirect path to track use compliance, but no significant direct path.

The results indicate that psychological distance could be used to aid compliance with boot cleaning and track use guidelines in kauri forests. Psychological distance showed significant indirect effects, and thus partial mediation, between the two of the three criterion variables of NEP score and trust in government with both dependent variables of compliance to boot cleaning and track use guidelines. Although physical distance from an infected kauri, as mentioned prior, was partially mediated in the boot cleaning compliance model, but fully mediated in the track use compliance model. This indicates that physical distance from an infected kauri may not impact track use compliance, except where it lowers psychological distance, which could be read that irrespective of how far away participants live they are reporting similar track use

compliance. However, it's important to note that the predictors in the model only explained a modest amount of variance in the dependent variables: 11% for boot cleaning, and 16% for track use.

This low amount of total variance is also an interesting issue. For one it indicates that none of the measured variables explain much of the exhibited behaviours. Yet, this small amount of explained variance in boot cleaning and track use can carry some clinical significance due to the hundreds of thousands of potential visitors to kauri forests yearly, meaning even small effect sizes can have large real world impacts. There are a couple potential reasons for this low explained variance, one being the high averages for each self-reported behaviour may mean that real world behavioural variance is lost. If this were the case, we may expect the explained variance of psychological distance on real world behaviours to be higher. Another reason may be that our chosen variables are simply not that important for these behaviours, perhaps contextual prompts, such as boot cleaning signage recommendations or requests are more important to self-reported or exhibited behaviours than any of the psychological components or physical distance.

## Theoretical implications

This study helps build the literature on the usefulness of psychological distance as a tool for encouraging PEB. However, it falls short of previous studies, such as Jones, Hine [17], through the inability to find a multidimensional psychological distance variable, and a relatively weak effect size of psychological distance on both boot cleaning and track use compliance. This indicates that although psychological distance is a component in PEB, it may be more powerful in generating intentions to engage in PEB [17], rather than translating those intentions into actual behaviours (or self-reported behaviours, as in the current study).

The study also indicates that exploring environmental worldview, through the NEP scale [23], and although only approaching scale reliability, is still appropriate for exploring pro-environmental behaviours in research. This was especially appropriate for track use, on which NEP had the strongest direct effect of this study. This possibly indicates that encouraging a pro-environmental worldview would be useful in reducing unwanted behaviour. What may cause hesitation in accepting this claim is the scale reliability issue, which although falling only slightly below the benchmark, it would add confidence to this interpretation if greater confidence can be given that the NEP scale can record environmental worldview accurately within the New Zealand context.

As found in prior studies, trust is important for participant receptiveness of communicated pro-environmental messages [31,35,41,42]. Distrust can also act as a barrier between interventions and the desired behaviour. This may be especially true for interventions requesting additional behaviour, over interventions to reduce behaviours. Specifically, while a pro-environmental worldview may aid in reducing unwanted behaviours, it may take trust that government has implemented guidelines benevolently for individuals to exhibit the sought PEB, rather than a more individually preferable, and potentially more environmentally destructive, alternative.

Physical distance showed a direct path with only boot cleaning. Future studies should aim to be highly specific about which PEB is of interest, as a general approach may lack the specificity to determine the intervenable mechanisms for a target behaviour. For example, if compliance behaviour was examined as a single variable, the differences between boot cleaning and track use would be missed, and an intervention developed due to any findings may not be as effective.

The predictor variables were also weakly (but significantly) correlated with each other. With the NEP scores being negatively correlated with both trust and physical distance. As for trust being negatively correlated with NEP, this could possibly indicate a lack of trust in government by the more environmentally minded. Reasons for this are currently unknown, but perhaps controversies around government inactions in terms of climate issues could explain this divergence. NEP was also negatively correlated with physical distance from an infected kauri tree, a possible explanation being that the closest participants (including all recruited from Titirangi) live in or very close to kauri forests, and thus those who choose to live in a forest setting may be more environmentally minded.

Trust was also positively correlated with physical distance, this could be due to the trust questions being based on the local governments for the kauri forests in question (both in the Auckland region). Trust was found to increase with greater

distance, this could possibly be due to knowing less about the issues and controversy of kauri dieback management or any impacts of mitigation becoming less personally relevant through increased physical distance.

## Practical implications

Future kauri dieback behaviour change interventions may wish to consider a multi-pronged approach. Although consistent with other studies [43], considering the weak to moderate effect sizes and the low amount of variance accounted for by our predictor variables, designing an intervention that includes multiple factors could improve its effectiveness. A multi-pronged intervention should seek to increase trust in government and encourage a more pro-environmental worldview, which in turn would potentially have both a direct effect on PEBs and an indirect effect through decreasing psychological distance. A prior qualitative study in the Waitakere ranges indicated participants held some distrust of the local government, some indicating they felt some government actions were diametrically opposed to the mitigation strategies imposed on forest users. For example, not allowing users to access the forest by track closures to protect kauri, but also destroying a block of kauri to build a new water pumping station [35,42]. As mentioned previously, trust is based on the truster feeling that the trustee is acting in their best interest. Not admitting possible hypocrisy is unlikely to engender faith that the government is working in their constituent's best interest. With such a large population as a city it is also likely that governing bodies are making some smaller groups of people sacrifice more for the benefit of the others, such as the locals who must host the pumping station for the benefit of Auckland city as a whole. Taking this into consideration, governments should advertise the benefits of their actions without hiding or obfuscating the detriments and sacrifices required, and should aim to approach, listen to, and mitigate the concerns of those asked to bare the greater sacrifices, while explaining why those sacrifices are necessary. Understanding that kauri forest locals will be baring a greater sacrifice to protect kauri due to the tracks being part of their daily life would be important to engendering greater trust. The local government did run local events and outreach for this purpose, but perhaps it was insufficient to protect trust, or perhaps misdirected at explaining the environmental impact of kauri dieback, rather than addressing the likely individual impacts, such as the required sacrifices.

The effect of psychological distance on both compliance behaviours may be too weak to warrant a psychological distance reducing intervention, but the incorporation of psychological distance reducing messages into other interventions may improve effectiveness. It should be noted that with hundreds of thousands of visitors to kauri forests every year a small effect can have a large impact on slowing the spread of kauri dieback.

Finally, physical distance to an infected kauri has a complex relationship with boot cleaning and track use compliance. Considering only the results of the path analysis, the direct negative relationship between physical distance and boot cleaning was weak and was not present with track use compliance. This may bode well for future interventions, as place of residence may not act as a significant barrier to PEB. To combat the weak indirect effect of physical distance, designing messages aimed to reduce the psychological distance of infrequent visitors to kauri forests may help to improve compliance.

## Limitations

The primary limitation is that this study employed a cross-sectional correlational research design, and although the results from the path analysis indicate a causal pathway operating through psychological distance, strong causal inferences cannot be made.

Another limitation of this study was an inability to find a multidimensional psychological distance variable. Unfortunately, due to the unidimensional factor which dimensions of psychological distance is affected by physical distance, trust in government, and pro-environmental worldview; along with how these separate dimensions impact the behaviours of interest, is unable to be investigated. Despite this, psychological distance shows some promise for kauri dieback mitigation interventions, but that it is likely unwise to base said intervention off psychological distance theories.

The NEP scale employed only approached reliability, which has limited the insights able to be gleaned from using this scale. Due to NEP reliability approaching our cronbach's alpha benchmark of.7, it was still employed under the couched

term of NEP score rather than pro-environmental worldview. The Trust in Government scale had a similar reliability issue, and a similar solution of referring it by the scale name, rather than referring to it as the concept of trust. Future studies may wish to find a more New Zealand relevant indicator of pro-environmental worldview and Trust in Government.

Along with scale reliability there were normality assumption violations. These violations are likely to have had only a minor impact on the analyses, as indicated by there being no statistical difference in track use pathway analysis model fit if using the robust maximum likelihood estimation over the maximum likelihood estimation. Despite this, a conservative reading of the results is recommended, but an understanding that most environmental psychological research will contain similar issues and that the findings should not be completely discounted due to them, is also recommended.

The final limitation is that self-reported behaviour may or may not map accurately onto actual exhibited behaviours, more about this in future research.

### Future research

Psychological distance had a weak effect on both boot cleaning and track use compliance, it would be expected that compliance would be stronger if psychological distance was a major driver to PEB. Due to this, it could be hypothesised that low psychological distance is simply one of many motivations for behaviour. Further research into how psychological distance may fit into other theories of environmental behaviour, such as capability, opportunity and motivation [44,45], may benefit both theories.

As mentioned above, both the NEP scale and Trust in Government scale had low reliability. This is in line with other NEP research in the New Zealand context [37]. Future research should be conducted to both determine why these scales reliability suffer with New Zealand participants and to develop more reliable scales for the New Zealand population.

Due to this study employing a self-report measure for behaviour, rather than a recording of actual behaviour another potential avenue for research is how a self-report for behaviour measured in hindsight matches exhibited behaviour in a real-world setting. Understanding the difference between self-report and actual behaviour would allow a better translation of these results into a real-world intervention. A recent study has found real-world boot cleaning partial to full compliance to be around 80%, with about 20% of that being full compliance [46]. As a rough comparison and a side note, full compliance within our study was self-reported at just under 50%, with partial compliance around 75%. This indicates, as expected, self-report is generally higher than real world behaviour. A study would be required to confirm these numbers, perhaps one which links participant self-reports with video recorded behaviours.

### Conclusion

Consistent with our hypothesis, psychological distance predicted compliance with both boot cleaning and track use guidelines, with decreasing psychological distance increasing compliance. Psychological distance was the strongest predictor of boot cleaning compliance, but was weaker than NEP score on track use compliance. Although the effect sizes were generally modest, hundreds of thousands of visitors will continue to use kauri forests every year, and the practical impact of a minor effect could accumulate to slow the spread of kauri dieback.

Meanwhile, psychological distance may add motivation to exhibit pro-environmental behaviour, but is unlikely to be the driving force. Exploring psychological distance as a component within another model, may be fertile ground for increasing future behavioural intervention effectiveness.

### Supporting information

**S1 Appendix. File is a table with the factor loadings of the Psychological Distance components and ultimate single factor Cronbach's Alpha.**
(DOCX)

## Acknowledgments

This article is part of the first author's Master of Science in Psychology thesis from the University of Canterbury, New Zealand. Thanks are extended to the senior supervisor Don Hine, and co-supervisors Andrea Grant and Nicole Lindsay, all who contributed to the research process. Prof. Don Hine who was guided survey development, the writing process and gave feedback for the thesis. Dr. Andrea Grant and Dr. Nicole Lindsay were both influential in survey development, and gave feedback to convert the thesis into this article.

## Author contributions

**Conceptualization:** Hugh A. N. Benson, Donald Hine.

**Data curation:** Hugh A. N. Benson.

**Formal analysis:** Hugh A. N. Benson, Donald Hine.

**Funding acquisition:** Andrea Grant.

**Investigation:** Hugh A. N. Benson.

**Methodology:** Hugh A. N. Benson, Donald Hine.

**Project administration:** Andrea Grant, Nicole Lindsay.

**Supervision:** Donald Hine, Andrea Grant, Nicole Lindsay.

**Visualization:** Andrea Grant, Nicole Lindsay.

**Writing – original draft:** Hugh A. N. Benson.

**Writing – review & editing:** Hugh A. N. Benson, Donald Hine, Andrea Grant, Nicole Lindsay.

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
