## [Decision Letter · Decision Letter 0]

7 May 2025

Dear Dr. Benson,

Thank you for submitting your manuscript to PLOS ONE. After careful consideration, we feel that it has merit but does not fully meet PLOS ONE’s publication criteria as it currently stands. Therefore, we invite you to submit a revised version of the manuscript that addresses the points raised during the review process.

We look forward to receiving your revised manuscript.

Kind regards,

Jianpeng Fan

Academic Editor

PLOS ONE

**Journal Requirements:**

1. When submitting your revision, we need you to address these additional requirements. Please ensure that your manuscript meets PLOS ONE's style requirements, including those for file naming. The PLOS ONE style templates can be found at https://journals.plos.org/plosone/s/file?id=wjVg/PLOSOne_formatting_sample_main_body.pdf and https://journals.plos.org/plosone/s/file?id=ba62/PLOSOne_formatting_sample_title_authors_affiliations.pdf 2. Thank you for stating the following financial disclosure: Master's Scholarship awarded to Hugh Benson which enabled him to conduct this research.Scholarship was awarded by Scion Research (https://www.scionresearch.com/) the New Zealand Crown Research Institute for Forestry. Scion had no input on methodology or results. One of the co-writers, Andrea Grant, is a Scion employee. Her primary role was as an editor, in which she suggested edits. No suggestions were made around changing results, nor would have been considered.A copy of these edit suggestions have been kept, and could be made available upon a reasonable request. Please state what role the funders took in the study.  If the funders had no role, please state: "The funders had no role in study design, data collection and analysis, decision to publish, or preparation of the manuscript." If this statement is not correct you must amend it as needed. Please include this amended Role of Funder statement in your cover letter; we will change the online submission form on your behalf. 3. Thank you for stating the following in the Acknowledgments Section of your manuscript: This article is part of the first author’s Master of Science in Psychology thesis from the University of Canterbury, New Zealand. Thanks are extended to the senior supervisor Don Hine, and co-supervisors Andrea Grant and Nicole Lindsay, all who contributed to the research process. Prof. Don Hine who was guided survey development, the writing process and gave feedback for the thesis. Dr. Andrea Grant and Dr. Nicole Lindsay were both influential in survey development, and gave feedback to convert the thesis into this article. Gratitude is also extended to the CRI for forestry Scion, for providing a Master’s Scholarship, which enabled the thesis project. We note that you have provided funding information that is not currently declared in your Funding Statement. However, funding information should not appear in the Acknowledgments section or other areas of your manuscript. We will only publish funding information present in the Funding Statement section of the online submission form. Please remove any funding-related text from the manuscript and let us know how you would like to update your Funding Statement. Currently, your Funding Statement reads as follows: Master's Scholarship awarded to Hugh Benson which enabled him to conduct this research.Scholarship was awarded by Scion Research (https://www.scionresearch.com/) the New Zealand Crown Research Institute for Forestry. Scion had no input on methodology or results. One of the co-writers, Andrea Grant, is a Scion employee. Her primary role was as an editor, in which she suggested edits. No suggestions were made around changing results, nor would have been considered.A copy of these edit suggestions have been kept, and could be made available upon a reasonable request. Please include your amended statements within your cover letter; we will change the online submission form on your behalf. 4. Thank you for stating the following in the Competing Interests section: I have read the journal's policy and the author of this manuscript have the following competing interest: Hugh Benson was awarded a Master's Scholarship whcih enabled him to conduct this research. The scholarship awarder (Scion Research) had no input into methodology or reported results. We note that one or more of the authors are employed by a commercial company.  a. Please provide an amended Funding Statement declaring this commercial affiliation, as well as a statement regarding the Role of Funders in your study. If the funding organization did not play a role in the study design, data collection and analysis, decision to publish, or preparation of the manuscript and only provided financial support in the form of authors' salaries and/or research materials, please review your statements relating to the author contributions, and ensure you have specifically and accurately indicated the role(s) that these authors had in your study. You can update author roles in the Author Contributions section of the online submission form. Please also include the following statement within your amended Funding Statement. “The funder provided support in the form of salaries for authors [insert relevant initials], but did not have any additional role in the study design, data collection and analysis, decision to publish, or preparation of the manuscript. The specific roles of these authors are articulated in the ‘author contributions’ section.”If your commercial affiliation did play a role in your study, please state and explain this role within your updated Funding Statement.  b. Please also provide an updated Competing Interests Statement declaring this commercial affiliation along with any other relevant declarations relating to employment, consultancy, patents, products in development, or marketed products, etc.   Within your Competing Interests Statement, please confirm that this commercial affiliation does not alter your adherence to all PLOS ONE policies on sharing data and materials by including the following statement: "This does not alter our adherence to  PLOS ONE policies on sharing data and materials.” (as detailed online in our guide for authors http://journals.plos.org/plosone/s/competing-interests) . If this adherence statement is not accurate and  there are restrictions on sharing of data and/or materials, please state these. Please note that we cannot proceed with consideration of your article until this information has been declared. Please include both an updated Funding Statement and Competing Interests Statement in your cover letter. We will change the online submission form on your behalf. 5. Please note that your Data Availability Statement is currently missing the direct link to access each database. If your manuscript is accepted for publication, you will be asked to provide these details on a very short timeline. We therefore suggest that you provide this information now, though we will not hold up the peer review process if you are unable. 6. Your ethics statement should only appear in the Methods section of your manuscript. If your ethics statement is written in any section besides the Methods, please delete it from any other section. 7. We note that this data set consists of interview transcripts. Can you please confirm that all participants gave consent for interview transcript to be published? If they DID provide consent for these transcripts to be published, please also confirm that the transcripts do not contain any potentially identifying information (or let us know if the participants consented to having their personal details published and made publicly available). We consider the following details to be identifying information:- Names, nicknames, and initials- Age more specific than round numbers- GPS coordinates, physical addresses, IP addresses, email addresses- Information in small sample sizes (e.g. 40 students from X class in X year at X university)- Specific dates (e.g. visit dates, interview dates)- ID numbers Or, if the participants DID NOT provide consent for these transcripts to be published:- Provide a de-identified version of the data or excerpts of interview responses- Provide information regarding how these transcripts can be accessed by researchers who meet the criteria for access to confidential data, including:a) the grounds for restrictionb) the name of the ethics committee, Institutional Review Board, or third-party organization that is imposing sharing restrictions on the datac) a non-author, institutional point of contact that is able to field data access queries, in the interest of maintaining long-term data accessibility.d) Any relevant data set names, URLs, DOIs, etc. that an independent researcher would need in order to request your minimal data set. For further information on sharing data that contains sensitive participant information, please see: https://journals.plos.org/plosone/s/data-availability#loc-human-research-participant-data-and-other-sensitive-data If there are ethical, legal, or third-party restrictions upon your dataset, you must provide all of the following details (https://journals.plos.org/plosone/s/data-availability#loc-acceptable-data-access-restrictions):a) A complete description of the datasetb) The nature of the restrictions upon the data (ethical, legal, or owned by a third party) and the reasoning behind themc) The full name of the body imposing the restrictions upon your dataset (ethics committee, institution, data access committee, etc)d) If the data are owned by a third party, confirmation of whether the authors received any special privileges in accessing the data that other researchers would not havee) Direct, non-author contact information (preferably email) for the body imposing the restrictions upon the data, to which data access requests can be sent?

Reviewers' comments:

Reviewer's Responses to Questions

**Comments to the Author**

1. Is the manuscript technically sound, and do the data support the conclusions?

Reviewer #1: Yes

Reviewer #2: Yes

2. Has the statistical analysis been performed appropriately and rigorously?

Reviewer #1: No

Reviewer #2: N/A

3. Have the authors made all data underlying the findings in their manuscript fully available?

Reviewer #1: No

Reviewer #2: Yes

4. Is the manuscript presented in an intelligible fashion and written in standard English?

Reviewer #1: Yes

Reviewer #2: Yes

**Reviewer #1: ** Initially, I’d like to congratulate the authors on the manuscript written. It has a good contextualization of the problem, the psychological variables that are used and the hypothesis of the study. I strongly recommend the publication of this article on PLOS One journal.

Nevertheless, I have some minor comments that may help the authors to enhance the quality of the study.

The tables were added to the article as image files, it’s necessary to insert the tables correctly, drawing the table with the tools provided by the app. This is a crucial thing to the forward steps of the article processing.

The data were analyzed with IBM SPSS and AMOS package. I strongly suggest to use updated softwares to the usage of Structural Equation Models (SEM), such as R and/or JASP (free access). Furthermore, it’s necessary to test the assumptions of the statistical analysis, to the correlation analysis, it’s necessary to test for normality of the distribution of the numerical variables, and for SEM models, it’s always necessary to test the multivariate normality,that is, considering all the variables that were considered in the study, this assumption can be tested with Mardia’s test.

If the assumptions for the correlation are violated, you should use the Spearman’s or the Kendall’s coefficient. In SEM models, you should use the Weighted Least Squares (WLS) estimator to ordinal data (Confirmatory Factor Analysis models), or Maximum Likelihood Robust in case the authors use numerical data (as in Path analysis)

Another good addition to the article would be the evaluation of the model fit of the scales, considering measurement models by SEM, that is, the Confirmatory Factor Analysis (CFA). In the methods section, the authors shows some psychometric indicators, but it’s not clear if they’re using indicators tested in another studies or if those indicators were calculated with the samples.

Based on this, I strongly suggest the authors to perform psychometric analysis to make sure that the scales used in the article fits to the data, considering the context in which they’re being used. Just after that the author’s could calculate a general score of the scales to use in the path analysis.

**Reviewer #2: ** 1. This text is logically organized and easy to read and follow.

2. The rationale, procedure, and analysis are adequately articulated.

3. A. The study does not specify how participants from the Titirangi sample were recruited, raising concerns about potential selection bias.

4. Several limitations cannot be resolved at this stage; however, more detailed explanations and corrective strategies can be presented and discussed to guide future research.

a. The most significant issue is the low explained variance, which ranges between 11% and 16%. A more comprehensive discussion of additional relevant factors is necessary to account for this limitation.

b. The Cronbach's alpha for the NEP scale is 0.68, which is relatively low. How might future research explore alternative methods to enhance the reliability of this measure?

c. As the study relies on self-report data, there is a risk of misrepresenting actual behaviour. It is important to consider complementary methods that may help mitigate such inaccuracies.

d. Adapting the trust scale appears to lack validity in the specific context of kauri dieback, and using a logarithmic transformation requires a more robust justification.

**Do you want your identity to be public for this peer review?** For information about this choice, including consent withdrawal, please see our Privacy Policy

Reviewer #1: No

Reviewer #2: **Yes: ** Dr. Ali M. AL-Asadi

---

## [Author Response · Author response to Decision Letter 1]

17 Jul 2025

Reviewers Comments:

1. Is the manuscript technically sound, and do the data support the conclusions?

a. R#1 – Yes

b. R#2 – Yes

c. No changes needed.

2. Has the statistical analysis been performed appropriately and rigorously?

a. R#1 – No

b. R#2 – N/A

c. Need to investigate Reviewer #1’s suggestions.

3. Have the authors made all data underlying the findings in their manuscript fully available?

a. R#1 – No.

b. R#2 – Yes

c. See editors comments #5, need ethics permission first.

4. Is the manuscript presented in an intelligible fashion and written in standard English?

a. R#1 – Yes

b. R#2 – Yes

c. No changes needed.

Below is my response to Reviewers suggestions:

Peer Reviewer Comments:

Reviewer #1: Initially, I’d like to congratulate the authors on the manuscript written. It has a good contextualization of the problem, the psychological variables that are used and the hypothesis of the study. I strongly recommend the publication of this article on PLOS One journal.

Nevertheless, I have some minor comments that may help the authors to enhance the quality of the study.

R#1.1 - The tables were added to the article as image files, it’s necessary to insert the tables correctly, drawing the table with the tools provided by the app. This is a crucial thing to the forward steps of the article processing.

R#1.2 - The data were analyzed with IBM SPSS and AMOS package. I strongly suggest to use updated softwares to the usage of Structural Equation Models (SEM), such as R and/or JASP (free access). Furthermore, it’s necessary to test the assumptions of the statistical analysis, to the correlation analysis, it’s necessary to test for normality of the distribution of the numerical variables, and for SEM models, it’s always necessary to test the multivariate normality,that is, considering all the variables that were considered in the study, this assumption can be tested with Mardia’s test.

R#1.3 - If the assumptions for the correlation are violated, you should use the Spearman’s or the Kendall’s coefficient. In SEM models, you should use the Weighted Least Squares (WLS) estimator to ordinal data (Confirmatory Factor Analysis models), or Maximum Likelihood Robust in case the authors use numerical data (as in Path analysis)

R#1.4 - Another good addition to the article would be the evaluation of the model fit of the scales, considering measurement models by SEM, that is, the Confirmatory Factor Analysis (CFA). In the methods section, the authors shows some psychometric indicators, but it’s not clear if they’re using indicators tested in another studies or if those indicators were calculated with the samples.

R#1.5 - Based on this, I strongly suggest the authors to perform psychometric analysis to make sure that the scales used in the article fits to the data, considering the context in which they’re being used. Just after that the author’s could calculate a general score of the scales to use in the path analysis.

Reviewer #2: 1. This text is logically organized and easy to read and follow.

2. The rationale, procedure, and analysis are adequately articulated.

3. A. The study does not specify how participants from the Titirangi sample were recruited, raising concerns about potential selection bias.

4. Several limitations cannot be resolved at this stage; however, more detailed explanations and corrective strategies can be presented and discussed to guide future research.

a. The most significant issue is the low explained variance, which ranges between 11% and 16%. A more comprehensive discussion of additional relevant factors is necessary to account for this limitation.

b. The Cronbach's alpha for the NEP scale is 0.68, which is relatively low. How might future research explore alternative methods to enhance the reliability of this measure?

c. As the study relies on self-report data, there is a risk of misrepresenting actual behaviour. It is important to consider complementary methods that may help mitigate such inaccuracies.

d. Adapting the trust scale appears to lack validity in the specific context of kauri dieback, and using a logarithmic transformation requires a more robust justification.

Response to peer review comments:

Thank you both for the time you have taken to provide such useful comments. I really appreciate your attention.

Below I have addressed the minor changes requested, and I hope you find these suitable.

Reviewer #1

R#1.1 - All tables have been changed from my created images to Word tables.

R#1.2 – I have run the analysis through RStudio and the results have come out identical to AMOS. For the Boot Cleaning model it is not possible to get a Robust ML due to the model having no degrees of freedom, but I did get one for the Track Usage model and the model fits were practically identical, differing by a couple variables in their rounding error. Robust TLI and TLI were also nearly identical, with Robust being slightly stronger. Due to this, I do not think it adds anything to change my software for this article. But I will switch to RStudio going forward as it does add functionality so thank you for the suggestion.

R#1.3 – I have again checked the violations of normality, and they were violated. I have investigated and taken your suggestion of using Spearman’s rho over Pearson’s r. It made only one difference in Trust was no longer correlated with boot cleaning compliance. I have changed all findings to reflect this.

R#1.4/5 – I agree that showing the model fit of the scales would be nice, but because one model is fully saturated and the other model has only a single degree of freedom it is a little pointless in this case. Previously, I did consider showing the growth of the model, but it adds words without adding ultimate meaning, so it was removed. I will hopefully have another article out which I do need to build the best model in this same instance, and for that the model fit indices are interesting and relevant. Thank you for the note though.

R#1.4/5 – Psychometric indicators: Thank you for this question. These Cronbach’s α were derived from this study’s sample. I have added text to that effect when describing the internal reliability.

Reviewer #2

R#2.3 – The methods employed were a letter drop down random streets and using facebook pages for the community of interest (Titirangi Community page). There is always a chance that it is not fully representative, but the need to balance confidentiality constraints and study feasibility does mean this is a risk that needs to be taken. I am overt at how this sample differs from the NZ public by looking at the demographics table (Table 1). I tried to do a really clear job to show how my sample differs. Some of these issues are due to Titirangi not generally matching the general NZ population (e.g. Higher level of Europeans vs other ethnicities), but we do the best we can and I do not believe any unnecessary bias has been included, nor that this study is unusually different to any other similar studies in participant recruitment. Thank you for your feedback though, it did give me a good chance to reflect for next time. I have added more details around recruitment method.

R#2.4 – I have attempted to add more detail to limitations and future research, primarily based on the below suggestions.

R#2.a – I agree with this comment and have expanded on potential reasons, while also mentioning that a small amount of variance for hundreds of thousands of forest visitors can still have a large clinical effect.

R#2.b – This measure has been difficult in the NZ context. Balador et al (2021) showed that it did not work that well, but unfortunately, since this was Master’s research the best I was able to do was validate his low reliability in the kauri dieback context. I have mentioned this in future research and limitations.

R#2.c – I agree. Luckily, real world compliance appears to be about 80%, with 20% being full compliance. This shows we are not too distant, but that more granularity is needed in our self-report measure to become more precise. I have included this study in the ‘future research’ section.

R#2.d – Unfortunately, Trust scales currently do not work very well in many environmental contexts. Personally, I believe trust is not well understood as a concept in environmental psychology, and I agree that this scale is less than ideal. We adapted this scale fairly directly and we got a similar result to Kettle and Dow. In future, I hope a study about trust in the kauri dieback context is conducted, but do feel this measure performed similar and has a similar validity to other trust measures employed in other studies. I agree that it is far from perfect, but do feel this reaches ‘good enough’ to add insight into the low impact of trust in psychological distance in the kauri dieback context. I have added some additional text stating how the measure was adapted by the research team to be about kauri dieback, as I agree that the original Kettle and Dow questions would not be appropriate if copied directly.

Greater justification has been provided for the distance logarithmic conversion:

A logarithmic conversion allows the linear measure of distance to be treated exponentially for impact on participants psychological distance. Allowing an overweighing in degrees of nearness, such as someone who lives next to an infected kauri tree, thus experiences this issue daily, may be impacted much more severely than another person who lives a single postcode away. While under weighing the linear impact of increasing distance, for example this same linear distance of a single postcode for participants living hundreds of kilometres away from an infected kauri will make very little difference in level of psychological distance impact.

Again, thank you both for your time and attention.

Regards,

Hugh Benson

---

## [Decision Letter · Decision Letter 1]

10 Sep 2025

Dear Dr. Benson,

Thank you for submitting your manuscript to PLOS ONE. After careful consideration, we feel that it has merit but does not fully meet PLOS ONE’s publication criteria as it currently stands. Therefore, we invite you to submit a revised version of the manuscript that addresses the points raised during the review process.

We look forward to receiving your revised manuscript.

Kind regards,

Jianpeng Fan

Academic Editor

PLOS ONE

Journal Requirements:

Reviewer's Responses to Questions

**Comments to the Author**

Reviewer #2: All comments have been addressed

Reviewer #3: All comments have been addressed

2. Is the manuscript technically sound, and do the data support the conclusions?

Reviewer #2: Yes

Reviewer #3: Yes

3. Has the statistical analysis been performed appropriately and rigorously?

Reviewer #2: Yes

Reviewer #3: Yes

4. Have the authors made all data underlying the findings in their manuscript fully available?

Reviewer #2: Yes

Reviewer #3: Yes

5. Is the manuscript presented in an intelligible fashion and written in standard English?

Reviewer #2: Yes

Reviewer #3: Yes

Reviewer #2: I thank the author for taking the time to address all my comments adequately. I have no further comments.

Reviewer #3: no changes are needed. The authors have addressed the questions raised in the original article. Now the manuscript is more readable and comprehensive. It has now become more technically sound.

**Do you want your identity to be public for this peer review?** For information about this choice, including consent withdrawal, please see our Privacy Policy

Reviewer #2: **Yes: ** Dr. Ali M. AL-Asadi

Reviewer #3: **Yes: ** Gyanesh Kumar Tiwari

---

## [Author Response · Author response to Decision Letter 2]

1 Oct 2025

All reviewer comments have been actioned.

---

## [Editor Report · Decision Letter 2]

6 Oct 2025

Understanding the Role of Psychological Distance in Preventing the Spread of Kauri Dieback

PONE-D-25-06009R2

Dear Dr. Benson,

We’re pleased to inform you that your manuscript has been judged scientifically suitable for publication and will be formally accepted for publication once it meets all outstanding technical requirements.

Kind regards,

Jianpeng Fan

Academic Editor

PLOS ONE
---

## [Editor Report · Acceptance letter]

PONE-D-25-06009R2

PLOS ONE

Dear Dr. Benson,

I'm pleased to inform you that your manuscript has been deemed suitable for publication in PLOS ONE. Congratulations! Your manuscript is now being handed over to our production team.

Kind regards,

on behalf of

Dr. Jianpeng Fan

Academic Editor

PLOS ONE